# Transcutaneous Electrical Cranial-Auricular Acupoint Stimulation Modulating the Brain Functional Connectivity of Mild-to-Moderate Major Depressive Disorder: An fMRI Study Based on Independent Component Analysis

**DOI:** 10.3390/brainsci13020274

**Published:** 2023-02-06

**Authors:** Lifang Liao, Liulu Zhang, Jun Lv, Yingchun Liu, Jiliang Fang, Peijing Rong, Yong Liu

**Affiliations:** 1College of Integrated Traditional Chinese and Western Medicine, Southwest Medical University, Luzhou 646000, China; 2Department of Magnetic Resonance Imaging, Affiliated Chinese Traditional Medicine Hospital, Southwest Medical University, Luzhou 646000, China; 3Department of Comprehensive Internal Medicine, Affiliated Chinese Traditional Medicine Hospital, Southwest Medical University, Luzhou 646000, China; 4Guang’anmen Hospital, China Academy of Chinese Medical Sciences, Beijing 100053, China; 5Institute of Acupuncture and Moxibustion, China Academy of Chinese Medical Sciences, Beijing 100005, China

**Keywords:** depression, transcutaneous electrical cranial-auricular acupoint stimulation (TECAS), escitalopram, resting-state functional magnetic resonance imaging, MRI, independent component analysis (ICA)

## Abstract

Evidence has shown the roles of taVNS and TECS in improving depression but few studies have explored their synergistic effects on MDD. Therefore, the treatment responsivity and neurological effects of TECAS were investigated and compared to escitalopram, a commonly used medication for depression. Fifty patients with mild-to-moderate MDD (29 in the TECAS group and 21 in another) and 49 demographically matched healthy controls were recruited. After an eight-week treatment, the outcomes of TECAS and escitalopram were evaluated by the effective rate and reduction rate based on the Montgomery–Asberg Depression Rating Scale, Hamilton Depression Rating Scale, and Hamilton Anxiety Rating Scale. Altered brain networks were analyzed between pre- and post-treatment using independent component analysis. There was no significant difference in clinical scales between TECAS and escitalopram but these were significantly decreased after each treatment. Both treatments modulated connectivity of the default mode network (DMN), dorsal attention network (DAN), right frontoparietal network (RFPN), and primary visual network (PVN), and the decreased PVN–RFPN connectivity might be the common brain mechanism. However, there was increased DMN–RFPN and DMN–DAN connectivity after TECAS, while it decreased in escitalopram. In conclusion, TECAS could relieve symptoms of depression similarly to escitalopram but induces different changes in brain networks.

## 1. Introduction

Depression is a mental illness characterized by an at least two-week constant feeling of sadness and/or guilt, diminishing interest in activities, or accompanied by the associated daily and social function deficits. Moreover, some patients with serious major depressive disorder (MDD) have suicidal ideation or behavior, causing large public health and economic burden for society and patients’ families [1,2]. According to the World Health Organization, MDD was ranked as the third cause of burden of disease worldwide in 2008, and it is estimated to rank first by 2030 [3]. Therefore, screening, diagnosis, and treatment are of vital importance.

Independent component analysis (ICA) analyzes resting-state functional connectivity (FC) within networks or between networks based on a blind source separation algorithm rather than the FC of voxels [4,5,6]. Studies using ICA have explored abnormal intra- and inter-regional networks involved in mental illness. Altered FC within and between networks of the default mode network (DMN) has been seen in depression [7,8,9,10,11,12,13,14], and Cao et al. [15] showed that there was significantly decreased internetwork FC between the anterior DMN and the salience network (SN), as well as the right frontal-parietal network (RFPN) in the suicidal depressed patients. Saris et al. [11] found a negative correlation between social dysfunction among patients with MDD and the intrinsic connectivity of the DMN. Additionally, altered DMN connectivity in patients with depression may be related to impaired internal monitoring and emotional regulation ability [16]. Furthermore, abnormalities have been seen in resting-state networks such as the central executive network (CEN), SN, frontoparietal network (FPN), ventral attention network (VAN), visual network (VN), and auditory network (AN), shown using ICA in depressed patients [7,8,17,18]. ICA also provides a way to learn about the treatment mechanism for psychiatric illness. For example, Martens et al. [19] suggested that the treatment responsivity to escitalopram was related to increased internetwork FC between the RFPN and the posterior DMN, somatomotor network, and somatosensory association cortex. A study of electroconvulsive therapy [20] showed that it enhanced the connectivity of the left cognitive executive network with the left angular gyrus and left middle frontal gyrus as well as its within-network connectivity; in addition, increased intraregional connectivity in the posterior DMN was positively correlated with the improvement in psychomotor retardation. Transcranial magnetic stimulation can restore the abnormal FC between the DMN and the precuneus network, left CEN, and sensory–motor network, and the network changes are correlated with an improvement in depressive symptoms and suicidal ideation, separately [21]. A follow-up study [22] showed that cognitive behavioral therapy may enhance the function of the SN and rehabilitate the abnormal brain network mode. Furthermore, it concluded that the left ventromedial prefrontal cortex and right inferior parietal lobule could be used as biomarkers for cognitive behavioral therapy.

For mild-to-moderate MDD, initial treatments include either antidepressants or psychotherapy or a combination of both if necessary [23,24]. However, the patient’s adherence to second-generation antidepressants is low, as a result of adverse neurological and gastrointestinal effects [25,26]. In addition, anti-depressants may be associated with an increased risk of suicide in young patients [27]. Psychotherapy can improve the symptoms of depression but there are inconsistent results regarding its superiority in short-term and long-term relapse prevention. Moreover, patients are significantly dependent on therapist availability and patients’ age [23,28,29]. As for treatment-resistant depression, vagus nerve stimulation (VNS) was approved by the United States Food and Drug Administration in 2005; VNS is divided into conventional implantable VNS (iVNS) and transcutaneous auricular VNS (taVNS), the former being an invasive treatment requiring surgical implantation, while the latter, as a non-invasive method, requires electrodes placed on the associated skin surface. VNS is mostly considered an adjunct treatment in guidelines for depressive disorder treatment. Although long-term VNS trials have demonstrated a sustainable benefit for patients with depression, short-term responses to treatment require further study [30,31,32,33].

Studies have shown that traditional acupuncture and electrical acupuncture at the Yin-Tang (EX-HN3) and Bai-Hui (DU20) sites could improve the symptoms of depression [34,35,36]. However, this minimally invasive method requires patients to visit acupuncturists weekly and comes with risks of pain, infection, and subcutaneous hematoma. A study by Fang et al. [37] suggested that taVNS could significantly normalize the intrinsic connectivity of the DMN in MDD, and Rong et al. [38] showed the potential application of VNS treatment for MDD. Transcutaneous electrical cranial stimulation (TECS) is an additional promising therapy for post-traumatic stress disorder and comorbid depression, as well as for cancer-related fatigue [39,40]. Based on the above studies, a new non-invasive stimulation method combined taVNS [37,38], and TECS named transcutaneous electrical cranial-auricular acupoint stimulation (TECAS)was developed showing high patient adherence in the treatment of depression. Besides, this electrical therapy could avoid the abovementioned disadvantages by placing the electrodes on the skin surface during the treatment process. As a novel treatment for depression, it is able to simplify the treatment processes of TECS and taVNS and produces synergistic effects. In the present study, the preliminary efficacy of TECAS was assessed using clinical scales and compared to escitalopram, a commonly used medication for depression. ICA was carried out to explore what similarities and differences occur in the modulation of brain networks after TECAS and escitalopram treatments and whether the altered brain networks could be associated with the changes seen on clinical scales.

## 2. Materials and Methods

### 2.1. Participants

Fifty patients with mild-to-moderate MDD and 49 age- and sex-matched healthy controls (HCs) were recruited from the Affiliated Chinese Traditional Medicine Hospital, Southwest Medical University, Luzhou, China. Each subject was recruited after screening by psychiatrists and voluntarily provided informed consent prior to enrollment. All participants in this trial were right-handed and native speakers of Chinese. In this prospective, single-blind, non-random study, 29 patients and 21 patients were assigned to the TECAS group and the escitalopram group, respectively. All patients met the ICD-10 diagnosis standard for mild-to-moderate MDD (Figure 1). In addition, patients were recruited if they fulfilled all of the following criteria: (1) age 18–70; (2) treatment-naïve depressive disorder or no associated treatment for more than two weeks prior to treatment; (3) no serious suicidal ideation or behaviors; (4) no psychotic symptoms; and, (5) no current enrollment in similar studies. The exclusion criteria were: (1) alcohol or drug dependence; (2) other mental illness or depression combined with other disorders; (3) dementia or other cognitive disorders. The inclusion criteria for HCs were: (1) age 18–70; (2) no depressive disorder or other mental illness previously or currently; and, (3) no history of any mental illness in first-degree relatives. In addition to the above exclusion criteria, pregnancy, serious brain trauma, neurological disease, serious physical diseases (such as serious chronic diseases, acute diseases, infectious diseases, or malignant tumors), and contraindications for MRI were also the exclusion criteria for all groups. The researchers assessing treatment outcomes were blinded to the allocation of depressed patients and their treatment conditions. This study was approved by the local Research Ethical Committee (project approval number: YJ-KY2019052; approved on 4 July 2019) and performed in accordance with the Declaration of Helsinki. Meanwhile, the study protocol has been previously registered at the Chinese Clinical Trials (number: ChiCTR2000029109).

### 2.2. Clinical Scales

All patients were evaluated using the Montgomery–Asberg Depression Rating Scale (MADRS), Hamilton Depression Rating Scale (HAMD), and Hamilton Anxiety Rating Scale (HAMA) at baseline and after 8 weeks of treatment. MADRS and HAMD are both validated questionnaires for the severity and treatment responsivity of depression. The difference between them is that the MADRS is a self- or clinician-rated scale, with higher scores indicating more severe depression [41], while HAMD needs to be assessed by clinicians [42]. HAMA is a validated 14-item scale for the measurement of psychic and somatic anxiety [43]. MADRS, HAMD, and HAMA scores were assessed by psychiatrists or trained research assistants pre- and post-treatment, and, during treatment, patient condition was self-recorded or recorded by clinical psychiatrists and treatment technicians. Clinical responses to TECAS and escitalopram were evaluated using the effective rate (ER) and reduction rate (RR); the former is defined as a ≥50% reduction in MADRS, HAMD, and HAMA scores from baseline to 8-week treatment, while the latter is the ratio of the reduction scores of MADRS, HAMD, and HAMA at 8 weeks to the baseline measurements. Additionally, adverse events of both treatments during the intervention procedure for every patient were recorded using the Rating Scale for Side Effects (SERS). In addition, HCs were also assessed using the above clinical scales except for SERS.

### 2.3. Intervention Procedures

For patients who received TECAS treatment, the intervention procedure was two sessions per day for 8 consecutive weeks, using the SDZ-IIB (Hwato, Jiangsu, China) electronic acupuncture instrument in either the supine or sitting position. The cranial electrodes were placed at the Yin-Tang (EX-HN3) and Bai-Hui (DU20) acupoints, while the auricular electrodes were placed on the bilateral distribution area of the vagus nerve in the cavity and cymba of the auricular concha (Figure 2). Patients were treated at home twice daily as required (morning and evening were suggested, with the evening treatment 30 min before bedtime). With the consideration of previous studies about electro-acupuncture stimulation, VNS, and trigeminal nerve stimulation [44,45,46,47] and our clinical experience, the cranial and auricular electrodes stimulation of TECAS should last for 30 min each session with a disperse-dense wave and a frequency of 4/20 Hz (4 Hz for 5 s, 20 Hz for 10 s, alternating). The intensity was adjusted to be tolerable without pain. Prior to treatment, the study team members instructed participants on the usage of the device to ensure that they were familiar with it, and participants were informed that they could contact the researchers if there were any questions during the 8-week treatment.

For patients in the escitalopram group, oral administration of escitalopram tablets was prescribed by psychiatrists for 8 consecutive weeks. In the first week, the usual dosage was 5 mg daily, then increased to twice daily starting in the second week. Considering patient’s age, drug responsiveness, and the severity and course of the disease, psychiatrists increased or reduced the daily dose but the maximum dose was not more than 20 mg daily. Moreover, older patients (>65 years old) received doses by half, including the starting and maximum dosages.

### 2.4. Data Acquisition and Preprocessing of fMRI

The baseline resting-state fMRI data from 50 patients with mild-to-moderate MDD diagnosed by psychiatrists and 49 age- and sex-matched HCs were collected in the Department of Magnetic Resonance Imaging, Affiliated Traditional Chinese Medicine Hospital, Southwest Medical University, Luzhou, China. In addition, the fMRI data of 29 patients in the TECAS group and 21 patients in the escitalopram group were collected at baseline and the end of intervention.

All images were collected on a Siemens Skyra 3.0-Tesla scanner with a 16-channel radio frequency coil. Functional images were acquired through an echo-planar imaging sequence with an echo time of 30 ms, repetition time of 2000 ms, flip angle of 90°, slice thickness of 3.5 mm, field of view of 220 × 220 mm^2^, matrix size of 64 × 64, and a total scan time of 400 s; 200 volumes were scanned in total. In addition, three-dimensional T1-weighted imaging was obtained by a magnetization-prepared rapid gradient echo sequence with an echo time of 2.98 ms, repetition time of 2530 ms, flip angle of 7°, slice thickness of 1 mm, field of view of 256 × 256 mm^2^, acquisition matrix of 256 × 256, and 176 slices. During the scan time, subjects were instructed to keep their eyes closed, relax, and stay awake.

The preprocessing procedure of fMRI data included removing time points, slice timing, realignment, normalization, and spatial smoothing, which was accomplished by the restplus software package [48]. Firstly, in order to ensure steady-state longitudinal magnetization and exclude the influence of noise, the first 10 time points for each subject were removed. Secondly, slice timing was adjusted in the MRI data by removing time points to avoid differences caused by scan order. Thirdly, realignment was performed to evaluate and remove unnecessary head motion (any participants with translational or rotational motion higher than 3 mm or 3° were excluded). Additionally, images were spatially normalized to the standard Montreal Neurological Institute (MNI) space after realignment and then resampled to 3 × 3 × 3 mm voxels. Finally, spatial smoothing was performed using a Gaussian kernel with a full width at half-maximum of 8 mm.

### 2.5. Independent Component Analysis

Group spatial ICA was performed using the GIFT toolbox (version 4.0b; http://mialab.mrn.org/software/gift/index.html; accessed on 5 June 2022.) for MATLAB (MathWorks, Natick, MA, USA), based on blind source separation for group and single-subject fMRI data. Firstly, it used principal component analysis to reduce the data dimensionality for each participant and composed an aggregate dataset; it then selected the information maximization (Infomax) algorithm and repeated this 100 times. Moreover, back reconstruction of the spatial map and time series of each independent component was performed according to the aggregate components and dimension reduction results; then, a minimum description length algorithm was used to identify the number of spatially independent components. Finally, the process involved selecting interesting functional networks in two ways. One used the automated process in the GIFT software; the components with the highest correlation coefficient with the reference templates from the automated anatomical labeling atlas were chosen for further analysis. In addition, networks were selected visually based on matching templates of resting-state networks [49]. Furthermore, the selected networks were characterized by low-frequency fluctuations (0.01–0.1 Hz) and needed to exclude components localized in cerebrospinal fluid, white matter, or those with low correlation to gray matter.

In total, the number of independent components extracted from HCs and mild-to-moderate MDD, between pre- and post-TECAS, and pre- and post-escitalopram were 32, 48, and 34, respectively. Various resting state networks were used to investigate changes in intra- and inter-network connectivity: (1) between mild-to-moderate MDD and HCs, the DMN, precuneus network, primary/higher VN (PVN/HVN), anterior/posterior SN (ASN/PSN), dorsal/ventral AN (DAN/VAN), left/right FPN (LFPN/RFPN), cerebellum, pre-central gyrus, insular network, and sensory–motor network were selected; (2) between pre- and post-TECAS, 17 resting-state networks were selected, including DMN, DAN/VAN, PVN/HVN, ASN/PSN, LFPN/RFPN, ASN/PSN, sensory–motor network, cerebellum, AN, pre-central network, insular network, and precuneus network; (3) between pre- and post-escitalopram, the DMN, ASN/PSN, PVN/HVN, LFPN/RFPN, SMN, cerebellum, AN, pre-central network, insular network, and precuneus network were chosen.

### 2.6. Statistical Analysis

Demographic and clinic data were analyzed using SPSS software (v. 26.0l IBM Corp., Armonk, NY, USA). The Chi-squared test (χ^2^) was used for sex and education level comparisons. Two-sample *t*-tests were used for comparisons of age and clinical scales. *p*-values < 0.05 were considered statistically significant.

The differences in MADRS, HAMD, and HAMA scores at baseline between the TECAS and escitalopram groups were first analyzed using a two-sample *t*-test (*p* < 0.05). After 8-week consecutive treatment, the clinical responses to TECAS and escitalopram were evaluated using the ER and RR. Within each group, paired *t*-tests (*p* < 0.05) were used to analyze differences in MADRS, HAMA, and HAMD scores from baseline to 8-week treatment. Between the TECAS and escitalopram group, in addition, to compare the ERs of MADRS, HAMD, and HAMA scores, differences of RRs in MADRS, HAMD, and HAMA scores after treatment were analyzed using a two-sample *t*-test (*p* < 0.05). Any adverse events were recorded by trained researchers.

Group spatial ICA was carried out to analyze the abnormal intra- and inter-network connectivity in three steps: (1) one was performed between mild-to-moderate MDD and HCs, (2) another was evaluated between pre- and post-TECAS; and, (3) the third analyzed the difference between pre- and post-escitalopram. Intra-network differences were inferred using SPM 12.0 software (https://www.fil.ion.ucl.ac.uk/spm/software/spm12; accessed on 5 June 2022). Firstly, a one-sample *t*-test (*p* < 0.05, false discovery [FDR]-corrected) was performed on resting-state brain networks in each group (including fMRI data from HCs and mild-to-moderate MDD groups, and pre- and post-TECAS or escitalopram), and it was saved as a template. Then, combining masks of the two sets were calculated for comparison using restplus software [45]. A two-sample *t*-test (*p* < 0.05, FDR-corrected) was used to compare the different resting-state brain networks in the groups (HCs and mild-to-moderate MDD), using age, sex, and MADRS, HAMD, and HAMA scores as covariates. Two-factor mixed ANOVA was performed to compare four group differences within the DMN, DAN/VAN, LFPN/RFPN, PVN/HVN, ASN, cerebellum, AN, pre-central gyrus, and insular network, with the covariates of age and sex. Two-factor tests included group factors (TECAS or escitalopram) and time factors (baseline and week 8). The interaction and main effects were analyzed at a family wise error (FWE)-corrected *p*-value of 0.05 using SPM 12.0, and group comparations of associated networks was limited to the combined masks obtained from one-sample *t*-tests (*p* < 0.05, FDR correction); these were then calculated by restplus software. Paired *t*-tests (*p* < 0.05, FDR corrected) were used for post-hoc analyses to evaluate the above abnormal intra-network connectivity between pre- and post-treatment in each group (TECAS or escitalopram), separately. Between-network differences were evaluated using the Mancovan toolbox (version 1.0) within GIFT software. Functional network connectivity (FNC) was calculated following the procedure described by Jafri et al. [50], and the FDR approach was used for multiple comparison correction. Finally, multiple regression analysis was performed to evaluate significant correlations (*p* < 0.05, FDR correlation) between abnormal intra- and inter-network FC changes and clinical scales (MADRS, HAMD, and HAMA); age and sex were also used here as covariates.

## 3. Results

### 3.1. Clinical and Demographic Data

As illustrated in Table 1, there were no significant differences between the mild-to-moderate MDD and HCs in age and sex (all *p* > 0.05) but there was a significant difference in education level (*p* < 0.05). At the end of the study, five patients voluntarily dropped out before the 8-week evaluation (two in the TECAS group (one for poor efficacy; the other for device damage and does not want to continue the treatment because of skin damage) and three in the escitalopram group (all are for poor efficacy, and one of them took another depressive drug during the 8-week treatment)). As seen in Table 2, there were also no significant differences between the TECAS and the escitalopram groups in age, sex, or education level (*p* > 0.05).

### 3.2. Clinical Scales for the Patients in the TECAS and Escitalopram Data

As shown in Table 2, there were no significant differences in the MADRS, HAMD, and HAMA scores at baseline (all *p* > 0.05) between the TECAS and escitalopram groups. The ER of MADRS in the TECAS group was 44.4%, while that in the escitalopram group was 55.5%. The ER of HAMD was 33.3% in the TECAS group and 38.8% in the escitalopram group. As for the ER of HAMA, the TECAS group (44.4%) was higher than that in the escitalopram group (38.8%). Further, the RRs of the three clinical scores after eight-week treatments were evaluated between the TECAS and escitalopram groups, with no significant differences (all *p* > 0.05). Moreover, within the TECAS and escitalopram groups separately, there were significant reductions in MADRS, HAMD, and HAMA scores after 8 consecutive weeks of treatment (MADRS between pre- and post-TECAS, paired *t*-test, *p* < 0.001; others, two-sample *t*-test, *p* < 0.001 (as a result of the low correlation of clinical scores between pre- and post-therapy)). Finally, the SERS scores showed no significant difference between the groups (*p* > 0.05) and the side effects of TECAS mainly included skin damage, fatigue, headache, sudation, thirst, and constipation.

### 3.3. Changes in the Brain Networks and Correlation Analysis

On the intra-network level, one-sample *t*-tests (*p* < 0.05, FDR corrected, voxels > 50) were used to observe the selected networks, then calculated using restplus software to make the combined masks. There was only a significantly decreased intra-network connectivity of the DMN in the mild-to-moderate MDD compared with HCs (two-sample *t*-test, *p* < 0.05, FWE correction; Figure 3). Between pre- and post-TECAS or escitalopram treatment, significant interaction effects were only observed in the insular network (*p* < 0.05, FWE corrected). Furthermore, significant temporal main effects were observed in the insular network, DAN, LFPN, PVN, and cerebellum, and significant main effects were observed in the ASN, VAN, and AN. However, in post-hoc analysis, there were no significant differences for the above networks (*p* < 0.05, FDR corrected) between pre- and post-escitalopram or TECAS treatment.

On the inter-network level, there were no statistically significant differences (*p* < 0.05, FDR correction) regarding FNC between HCs and mild-to-moderate MDD, pre- and post- TECAS, and pre-and post-escitalopram groups. Therefore, abnormal inter-networks were reported at the significance level of *p* < 0.05, uncorrected (Figure 4, Figure 5 and Figure 6). Specifically, there was decreased DMN–DAN connectivity in mild-to-moderate MDD compared with HCs (*p* < 0.05, uncorrected), and DMN–DAN connectivity was increased in the post-TECAS data as compared to baseline (*p* < 0.05, uncorrected). The same analysis was performed between pre- and post-escitalopram, showing decreased DMN–DAN connectivity after its treatment (*p* < 0.05, uncorrected). Additionally, there was a significantly increased DMN–RFPN connectivity in mild-to-moderate MDD compared to HCs (*p* < 0.05, uncorrected). Between pre- and post-therapy, there was increased DMN–RFPN connectivity after TECAS treatment (*p* < 0.05, uncorrected), while there was decreased DMN–RFPN connectivity after escitalopram treatment (*p* < 0.05, uncorrected). Moreover, there was increased PVN–RFPN connectivity in mild-to-moderate MDD compared to HCs (*p* < 0.05, uncorrected), and after 8-week treatment with TECAS or escitalopram, PVN–RFPN connectivity both decreased compared with the baseline (*p* < 0.05, uncorrected). In the escitalopram group, there was also decreased DMN/ PVN–VAN connectivity and increased DAN–LFPN connectivity after its treatment. Finally, multiple regression analysis showed only a positive correlation between the altered intrinsic connectivity in the superior frontal gyrus, anterior cingulate gyrus, and HAMD and HAMA scores at baseline in mild-to-moderate MDD; however, within the TECAS or escitalopram group, there were no significant differences between the altered brain networks from baseline and ∆clinical scales (after intervention minus baseline) after the treatment.

## 4. Discussion

In this study, the effectiveness of TECAS was explored for the treatment of mild-to-moderate MDD, using resting-state fMRI to investigate its interventional brain effects on intra- and inter-brain network connectivity. The results suggested that TECAS could exert similar antidepressant effects in a safe manner compared to escitalopram. Both treatments modulated DMN, DAN, RFPN, and PVN, and decreased PVN–RFPN connectivity, which might be a common mechanism of them. DMN–RFPN and DMN–DAN connectivity increased after TECAS treatment, while that decreased in the escitalopram group. Furthermore, altered functional connectivity of VAN and LFPN was only seen in the escitalopram group, and DMN/PVN–VAN and DAN–LFPN connectivity were altered after its treatment. However, multiple regression analysis showed no significant differences between the altered brain networks from baseline and ∆ clinical scales (after intervention minus baseline) after eight-week daily TECAS or escitalopram treatment.

MADRS and HAMD are often jointly used to evaluate the severity of depression and assess anti-depressive effects, such as antidepressants, electroconvulsive therapy, and transcranial magnetic stimulation [51,52,53]. Specifically, the MADRS items are a supplement for the psychometric limitations of HAMD [54]. The HAMA is a validated 14-item scale for the measurement of psychic and somatic anxiety [43]. In this study, MADRS, HAMD, and HAMA scores in the TECAS and escitalopram groups were both reduced after treatment (*p* < 0.001). The RRs of the above three clinical scores after 8-week treatment showed no significant differences between the TECAS and escitalopram groups, suggesting that both can improve the related symptoms of depression and anxiety with similar efficacy. The results from a multi-center, randomized controlled, non-inferiority trial by Yang et al. [55] are in accordance with the findings of this study, demonstrating that TECAS has potential as an optional treatment for depression in the future. Furthermore, SERS is a clinical scale including 14 items, with higher scores meaning severer side effects; there was also no significant difference in SERS scores between the TECAS and escitalopram groups. The side effects of TECAS mainly included skin damage, fatigue, headache, sudation, thirst, and constipation.

The brain mechanism of TECAS in depression was analyzed using ICA on alteration of intra- and inter-network connectivity. In this study, various networks were involved, especially the DMN, DAN/VAN, LFPN/RFPN, and PVN. The DMN is a network characterized by a high degree of FC with spontaneous and virtually continuous functions, being attenuated only when engaging in goal-directed actions [56,57]. Patients with MDD showed increased intrinsic connectivity of the DMN (such as medial prefrontal cortex, subgenual anterior cingulate cortex, thalamus, and precuneus) and altered connectivity between the DMN and other brain regions (including DAN, FPN, central executive network, and SN) related to the symptoms of the illness [57,58,59]. However, in our study, there was a significantly decreased intrinsic connectivity of DMN in mild-to-moderate MDD compared with HCs (*p* < 0.05, FWE correction). The reasons may be associated with the severity of depression and individual differences of patients, and all patients are mild-to-moderate MDD in our study. Furthermore, there were no significant differences in intrinsic connectivity of the DMN and other networks between pre- and post-TECAS or escitalopram.

On the inter-network connectivity, the present study showed that mild-to-moderate MDD had decreased DMN–DAN connectivity and increased DMN–RFPN connectivity compared with the HCs. There was increased connectivity between DMN and DAN, and DMN and RFPN after eight-week treatment with TECAS compared with baseline, while there was decreased connectivity between DMN and DAN, and DMN and RFPN, after escitalopram treatment. Martens et al. [19] concluded that treatment responsivity to escitalopram was associated with enhanced resting-state FC between RFPN and the posterior DMN. Therefore, the therapeutic effects of TECAS in our study had some commonalities with the brain regions related to escitalopram efficacy found by Marten et al. [19]; this also suggests that the DMN is a necessary network that should be investigated for the therapeutic mechanisms of TECAS in depression in the future. The DAN is of vital importance in mediating goal-directed stimulus-response selection [60]. This network mainly includes the intraparietal sulcus, frontal eye fields, and ventral precentral and middle frontal gyri [61], and is also a part of the frontoparietal control system [60,62]. It was suggested that depression with less accurate memory had reduced FC between the DAN and DMN [63]. In the present study, after TECAS treatment, DMN–DAN connectivity increased from the baseline, showing that TECAS may play a role in attenuating the disruption of the control system and improving memory. After escitalopram treatment, there was increased DAN–LFPN connectivity as compared to baseline. Kaise et al. [58] suggested hypoconnectivity between the FPN and DAN in MDD, which involves the cognitive control of attention, emotion regulation, and attending to the external environment. Therefore, the neurological effects of escitalopram in mild-to-moderate MDD could be associated with DAN–LFPN connectivity. The LFPN and RFPN are both parts of the frontoparietal control network, activated by various cognitive tasks [62,64], mainly including the bilateral dorsolateral prefrontal cortex, inferior parietal lobe, and middle temporal gyrus [65]. The frontoparietal control network often acts as a mediator for the interaction of the brain networks [65,66]. Altered DMN–FPN connectivity is thought to underlie deficits in processing and regulating affective stimuli [67]. Therefore, TECAS could relieve the affective symptoms of depression. In addition, the mild-to-moderate MDD group without treatment showed increased PVN–RFPN connectivity compared to the HCs in the present study, and the increased PVN–RFPN connectivity decreased after 8-week treatment with TECAS and escitalopram. The PVN is a network mainly including the middle occipital gyrus, middle frontal gyrus, lingual gyrus, precuneus, precentral gyrus, postcentral gyrus, fusiform gyrus, and para-hippocampal gyrus, and plays a vital role in memory-related mental imagery and visual memory consolidation processes [68]. Therefore, memory-related symptoms in mild-to-moderate MDD might improve after both treatments, and decreased PVN–RFPN connectivity may be the common mechanism related to symptom improvement in TECAS and escitalopram treatment. Moreover, decreased DMN/PVN–VAN connectivity was only seen after escitalopram treatment. The VAN is a network modulated by emotional stimuli [69]. Patients with depression have shown stronger effective connectivity within the VAN, and changes in VAN connectivity in depression were associated with attention bias [70,71]. Therefore, this suggests that escitalopram could be associated with the management of emotional stimuli and improvement of attention bias in depression, which is different from TECAS.

## 5. Limitations

Some limitations in this study must be noted. Firstly, this is a preliminary and non-randomized trial (due to ethical concerns), as the first study to use TECAS alone on patients suffering from mild-to-moderate depression and Yang et al. study [52]. After finding that TECAS could significantly reduce depressive symptoms, we recruited a second cohort to test the treatment effect and brain mechanism of TECAS compared to escitalopram. Since the baseline characteristics of patients in the two cohorts were similar and the study was completed within 1 year, it could influence the validity of the study as little as possible. Nevertheless, a randomized clinical trial is needed further. In addition, the sample size was small, especially in the escitalopram group (only 18 patients completed treatment). Secondly, although there were at least 2 weeks without medication, for patients who were not medication-naïve before the study, it is difficult to explain whether this exerted an influence on our results. Thirdly, although the researchers provided instruction on TECAS device usage prior to intervention, and the performance was supervised by telephone daily during the treatment, it is difficult to ensure that each patient performed the procedure in a standardized manner at home. Finally, there was a significant difference in education level between HCs and mild-to-moderate MDD, which may have influenced the results. To eliminate the difference, Raven’s Standard Progressive Matrices were performed in them. Unfortunately, as a result of some reasons, only 36 HCs and 28 patients completed the test, and a two-sample *t*-test of their IQ scores was not significantly different. (HCs: 109.22 ± 14.15; the patients group: 103.93 ± 12.31; *p* = 0.121 > 0.05). It suggests that education level and IQ scores could be better collected together if possible in similar research.

## 6. Conclusions

TECAS could improve the symptoms of depression in a safe manner, showing similar efficacy to escitalopram. ICA showed there were some altered brain networks after the eight-week treatment of TECAS, and especially the decreased PVN–RFPN connectivity might be the common brain mechanism for TECAS and escitalopram. This study provides evidence of a new treatment method for depression and a new way to explore its brain mechanisms. Further randomized controlled studies with larger datasets should be carried out to evaluate its treatment efficacy and explore the related brain mechanisms furtherly.

## Figures and Tables

**Figure 1 brainsci-13-00274-f001:**
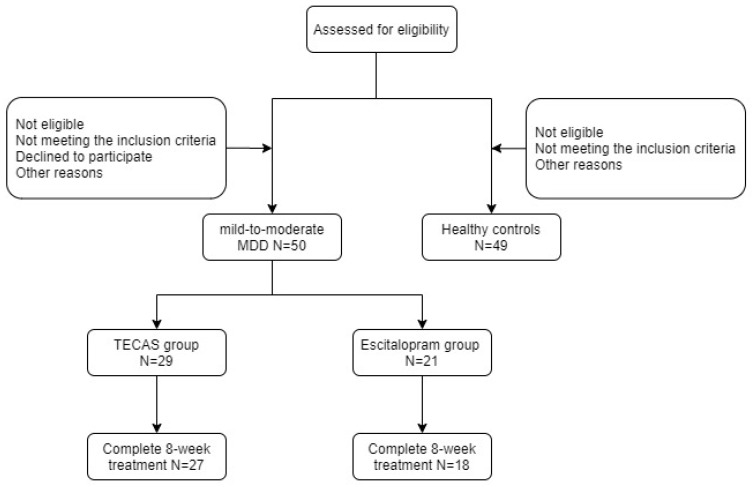
Study flow chart.

**Figure 2 brainsci-13-00274-f002:**
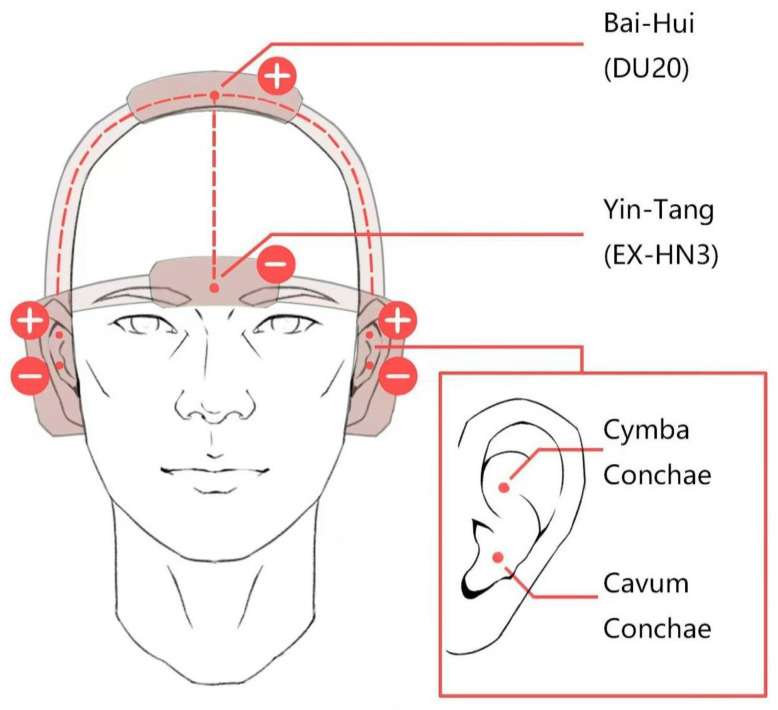
Illustration of transcutaneous electrical cranial-auricular acupoint stimulation (TECAS) “+” represents positive electrode, while “−” represents negative electrode.

**Figure 3 brainsci-13-00274-f003:**
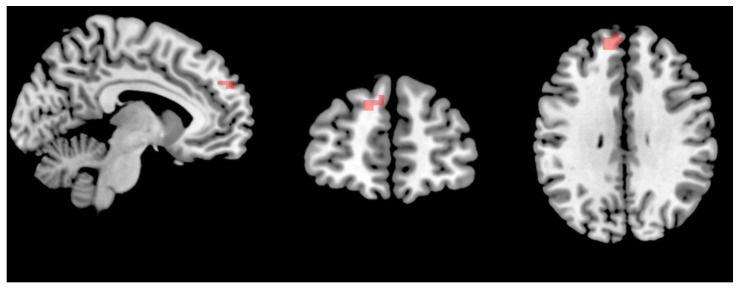
**Intra-network changes in MDD as compared to HCs.** The red-color brain region showed there was a decreased connectivity in MDD within the DMN (Frontal_Sup_Medial_L, AAL) compared with HCs (*p* < 0.05, FWE correction). Peak Montreal Neurological Institute atlas MNI coordinates: (−9 54 30).

**Figure 4 brainsci-13-00274-f004:**
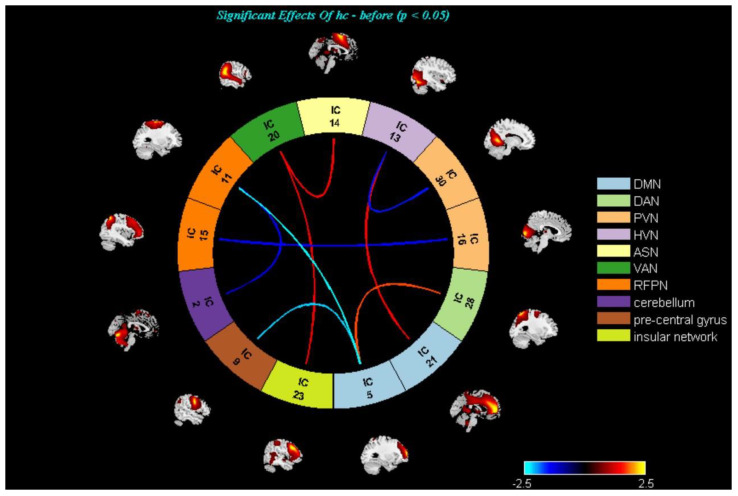
**MDD and HCs**; There was decreased DMN−DAN connectivity (*p* < 0.05, uncorrected), and increased DMN–RFPN and PVN–RFPN connectivity (*p* < 0.05, uncorrected) in the MDD compared to HCs. Warmer colors (positive values) represent a strengthened correlation between both groups, while cooler colors (negative values) represent a weakened correlation.

**Figure 5 brainsci-13-00274-f005:**
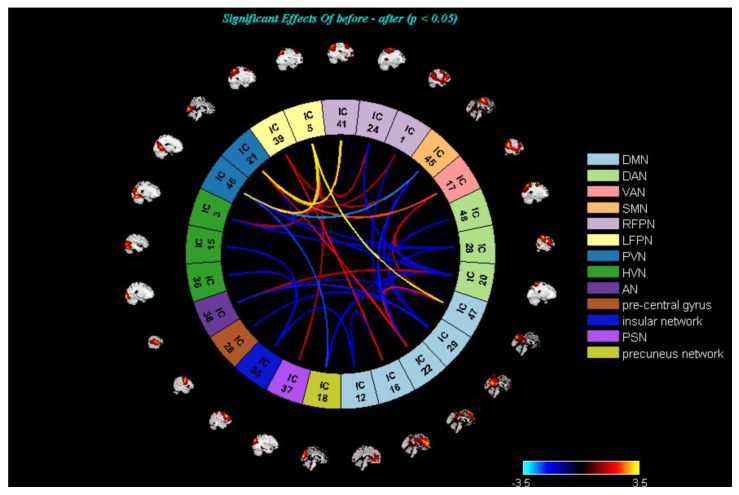
**TECAS group**; There were increased DMN−DAN and DMN−RFPN connectivity, and decreased PVN−RFPN connectivity after eight-week treatment (*p* < 0.05, uncorrected). Warmer colors (positive values) represent a strengthened correlation between pre– and post–TECAS treatment, while cooler colors (negative values) represent a weakened correlation.

**Figure 6 brainsci-13-00274-f006:**
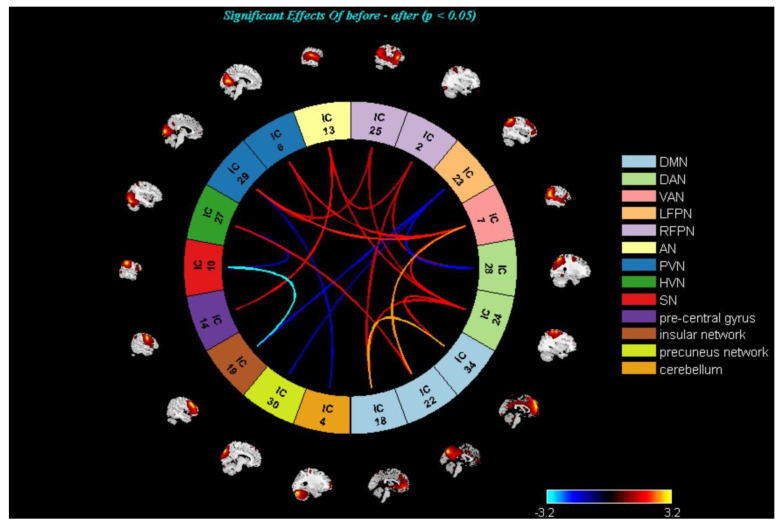
**Escitalopram group**; There were decreased DMN–RFPN, DMN–DAN/VAN, and PVN–VAN/RFPN connectivity but an increased DAN–LFPN connectivity after eight−week treatment (*p* < 0.05, uncorrected). Warmer colors (positive values) represent a strengthened correlation between pre– and post–escitalopram, while cooler colors (negative values) represent a weakened correlation.

**Table 1 brainsci-13-00274-t001:** Demographics and Clinical Characteristics of the Subject Groups.

Variables	HCs	MDD	Analysis	
			*p* Value	χ^2^
Sample size	49	50	-	-
Age (years) ^a^	46.57 ± 11.87	51.08 ± 11.77	0.061	-
Sex (M/F) ^b^	15/34	8/42	-	0.085
Education level (none/PS/JSH/CD) ^b^	0/17/9/23	10/13/13/14	-	0.01 *
MADRS ^a^	1.37 ± 1.67	16.48 ± 4.08	<0.001	-
HAMD ^a^	1.61 ± 2.35	13.70 ± 5.45	<0.001	-
HAMA ^a^	1.06 ± 1.53	13.44 ± 5.40	<0.001	-

PS, primary school; JSH, junior/senior high school; CD, college education; MDD, major depressive disorder; HCs, healthy controls. ^a^ Two-tailed *t*-test; ^b^ Pearson’s Chi-squared two-tailed test; * Fisher’s precision probability test.

**Table 2 brainsci-13-00274-t002:** Demographics and Treatment Response in Clinical Scales between Pre- and Post-TECAS or Escitalopram.

	TECAS			Escitalopram			
	Before	After	*p1* Value	Before	After	*p2* Value	*p3* Value
Sample size	27			18			
MADRS ^ac^	16.33 ± 3.69	9.00 ± 4.867	<0.001	16.67 ± 4.88	7.72 ± 3.89	<0.001	0.795 ^△^
HAMD ^ac^	16.41 ± 3.91	10.04 ± 4.10	<0.001	14.83 ± 3.33	9.11 ± 5.12	<0.001	0.168 ^△^
HAMA ^ac^	15.26 ± 4.71	9.44 ± 4.80	<0.001	15.22 ± 3.69	9.61 ± 4.85	<0.001	0.978 ^△^
ER in MADRS		44.40%			55.50%		
ER in HAMD		33.30%			38.80%		
ER in HAMA		44.40%			38.80%		
RR in MADRS ^a^		0.457 ± 0.246			0.526 ± 0.233		0.351
RR in HAMD ^a^		0.363 ± 0.261			0.360 ± 0.379		0.977
RR in HAMA ^a^		0.258 ± 0.652			0.335 ± 0.356		0.648
SERS ^a^		6.55 ± 3.17			5.44 ± 3.601		0.321
Age (years) ^a^		50.63 ± 12.61			51.5 ± 11.96		0.818
Gender (M/F) ^b^		6/21			2/16		0.445 *
Education (none/PS/JSH/CD) ^b^		5/7/4/11			5/7/4/11		0.119 *

*P1* Value, within TECAS group analysis; *p2* Value, within escitalopram group analysis; *p3* Value, analysis between TECAS group and escitalopram group; ER, effective rate; RR, reduction rate; PS, primary school; JSH, junior/senior high school; CD, college education; ^△^, the difference of baseline MADRS, HAMD, and HAMA scores; ^a^ two-tailed *t*-test; ^b^ Pearson’s Chi-squared two-tailed test; ^c^ two-tailed paired *t*-test; * Fisher’s precision probability test.

## Data Availability

Data can be made available upon reasonable request.

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
