# Peer review of "Transcutaneous Electrical Cranial-Auricular Acupoint Stimulation Modulating the Brain Functional Connectivity of Mild-to-Moderate Major Depressive Disorder: An fMRI Study Based on Independent Component Analysis"

_brainsci, 2023, doi:10.3390/brainsci13020274_

Round 1
Reviewer 1 Report
As far as the study is concerned, it has been well conducted. The results of the study are very interesting. I am concerned about the presentation of the results section because it does not follow the APA style and the Data Availability Statement. My minors changes are:
- line 40: It appears from what you wrote that two weeks are sufficient for a depression diagnosis
- line 117: How did the screening by psychiatrists work?
- line 123: "treatment-naïve depressive disorder or no associated treatment for more than two weeks prior to treatment". The effects of depression treatment usually take a long time to manifest. "no associated treatment for more than two weeks prior to treatment" does not exclude that a depression treatment received one year prior to the experiment could still affect the results.
-line: 165: "Patients were treated at home twice daily as required". Are the patients being treated by the researchers?
- 175: please provide more information about the 5 mg? why 5 mg? Why did you make that choice?
- 244-245: problem with the formatting
- 286 and following: the "p" value should'nt be in capitalized and no number before the dot "."
- 304 and following: the no sognificant results should be report in APA stye (" P > 0.05" is not following the APA style). Please format the results according to the APA style
- 318 and following: please re-write the p value according to the APA format
- 465: Maybe create a new sub title for the study limitations?
- 505: why don't make plublic your data? Wouldn't that make your paper stronger?
Author Response
line 40: It appears from what you wrote that two weeks are sufficient for a depression diagnosis.
- According to ICD-10, the standard course of depression were at least two weeks, and it was not seriously written in our article, which has been revised.
- line 117: How did the screening by psychiatrists’ work?
- Primary screening by clinical scales, such as Montgomery–Asberg Depression Rating Scale, Hamilton Depression Rating Scale, and Hamilton Anxiety Rating Scale.
- According to ICD-10, seriously obey to the diagnostic standard of mild and moderate depressive disorder.
- Revised and carried out the including and excluding criteria during the whole research.
- line 123: "treatment-naïve depressive disorder or no associated treatment for more than two weeks prior to treatment". The effects of depression treatment usually take a long time to manifest. "no associated treatment for more than two weeks prior to treatment" does not exclude that a depression treatment received one year prior to the experiment could still affect the results.
- In our study, most patients are mild-to-moderate treatment-naïve depressive disorder, and for the others, the treatment time and daily dosage are less than the serious major depressive disorder.
- Besides, in the limitation part, we have discussed this question. We would seriously consider it in our further study.
-line: 165: "Patients were treated at home twice daily as required". Are the patients being treated by the researchers?
- Patients were treated at home twice daily by themselves, but Prior to treatment, the study team members instructed participants on the usage of the device to ensure that they were familiar with it, and participants were informed that they could contact with the researchers if there were any questions during the 8-week treatment.
- 175: please provide more information about the 5 mg? why 5 mg? Why did you make that choice? - According to the drug instruction, the usual dosage was 10 mg daily, and considering the patients in our study were all mild-to-moderate depressive disorder and requiring suggestion from psychiatrists, the 5 mg daily was made in the first week, but twice daily from the second week.
- 244-245: problem with the formatting - has been formatted.
- 286 and following: the "p" value shouldn't be in capitalized and no number before the dot "." - has been revised.
- 304 and following: the no significant results should be report in APA stye (" P > 0.05" is not following the APA style). Please format the results according to the APA style.
- has been revised.
- 318 and following: please re-write the p value according to the APA format
- has been revised.
- 465: Maybe create a new subtitle for the study limitations?
- The limitation part has been inserted in our revised manuscript.
- 505: why don't make plublic your data? Wouldn't that make your paper stronger? - Firstly, we would be very happy to people who are interested in our research.
- Data can be made available upon reasonable request by corresponding author.
- Besides, I didn't know how to public data before submitting the manuscript.
Reviewer 2 Report
Firstly , I want to thank everyone working with this study, it's a great job
I just need some explanation for these scales used and over all paper need to be more concise
Author Response
"I just need some explanation for these scales used and over all paper need to be more concise".
- Montgomery–Asberg Depression Rating Scale (MADRS), Hamilton Depression Rating Scale (HAMD) are both validated questionnaires for the severity and treatment responsivity of depression. The difference between them is that the MADRS is a self- or clinician-rated scale, with higher scores indicating more severe depression [38], while HAMD needs to be assessed by clinicians [39]. HAMA is a validated 14-item scale for the measurement of psychic and somatic anxiety [40]. All patients were evaluated using MADRS, HAMD, and HAMA at baseline and after 8 weeks of treatment. Clinical responses to TECAS and escitalopram were evaluated using the effective rate (ER) and reduction rate (RR); the former is defined as a ≥50% reduction on MADRS, HAMD, and HAMA scores from baseline to 8-week treatment, while the latter is the ratio of the reduction scores of MADRS, HAMD and HAMA at 8 weeks to the baseline measurements.
- Besides, during the analysis of the fMRI data, these clinical scales were made as covariants.
The introduction part and the cite references have been improved as serious as possible.
Reviewer 3 Report
This is an interesting prospective single-blinded non-randomized study. The authors compared the use of transcutaneous electrical cranial-auricular acupoint stimulation with escitalopram as a treatment for major depressive disorder. They used clinical scales to compare the results of 8 weeks treatment. And fMRI to evaluate the changes in comparison to health controls.
The study is very interesting and novel. However, I believe some points could be clarified:
The non-randomization of the patients creates a bias for the efficacy of treatment that should be discussed. I believe the reason for select the patients to each of the groups should be explained.
The reasons for drop out and the side effects should be described for each of the participants. It is not clear why 5 patients voluntarily drop out and if there is any relation between the drop out and the side effects
The results section is bit difficult to follow in the segment regarding the fMRI results, it is not very clear why some of the “p-values” appear without correction. A table with a summary of the results would help to understand the comparisons.
Some minor points and suggestions include:
1. State in the abstract the difference in the clinical scale pre and post treatments.
2. Include a reference after the statement: “Independent component analysis (ICA) analyzes resting-state functional connectivity (FC) within networks or between networks based on a blind source separation algorithm rather than the FC of voxels”.
3. Some of the segments have big spaces in between words.
Author Response
The non-randomization of the patients creates a bias for the efficacy of treatment that should be discussed. I believe the reason for select the patients to each of the groups should be explained.
- We used this non-randomization strategy mainly due to ethical concerns. As the first study to use TECAS on patients suffering from mild and moderate depression and Yang et al ( https://doi.org/10.3389/fpsyt.2022.829932) study, we thought it would be wise to first test the effectiveness of TECAS. After demonstrating that TECAS could significantly reduce patients’ symptoms, we recruited a second cohort of patients to test the treatment effect and brain mechanism of TECAS compared to escitalopram. Since the baseline characteristics were similar in the two cohorts of patients and the study was completed within 1 year, we do not expect the design will influence the validity of the study. Nevertheless, a randomized clinical trial is needed in the future.
- As for the patients' selection of each group, we combined the patients' condition and psychiatrists' suggestion, and make sure the baseline characteristics were similar in the two cohorts of patients and the study was completed within 1 year.
The reasons for drop out and the side effects should be described for each of the participants. It is not clear why 5 patients voluntarily drop out and if there is any relation between the drop out and the side effects.
- Two in the TECAS group [One is poor efficacy; another is device damage, and don’t want to continue the treatment because of skin damage]
- Three in the escitalopram group [All are poor efficacy, and one of them took another depressive drug during the 8-week treatment.]
The results section is bit difficult to follow in the segment regarding the fMRI results, it is not very clear why some of the “p-values” appear without correction. A table with a summary of the results would help to understand the comparisons.
Thanks for your suggestion, it was a pity that there were no significant results with FWE/FDR correction, therefore we report the results without correction, and it is the primary exploration for the brain mechanism of TECAS, and we would carry out more associated research further. As for the table with summary of the results, we think the table would repeat the figures in the manuscript, and the figure with notes would be better to demonstrate the results than the table.
Some minor points and suggestions include:
State in the abstract the difference in the clinical scale pre and post treatments.
- The difference in the clinical scale pre and post treatments has been stated in the abstract.
Include a reference after the statement: “Independent component analysis (ICA) analyzes resting-state functional connectivity (FC) within networks or between networks based on a blind source separation algorithm rather than the FC of voxels”.
- Has been inserted.
- Calhoun VD, Adali T, Pearlson GD, Pekar JJ. 2001. A method for making group inferences from functional MRI data using independent component analysis. Hum Brain Mapp. 14:140–151
- Bell AJ, Sejnowski TJ. 1995. An information-maximization approach to blind separation and blind deconvolution. Neural Comput. 7:1129–1159
- Shahhosseini Y, Miranda MF. Functional Connectivity Methods and Their Applications in fMRI Data. Entropy (Basel). 2022 Mar 11;24(3):390. doi: 10.3390/e24030390. PMID: 35327901; PMCID: PMC8946919.
Some of the segments have big spaces in between words.
- Have been revised.